# An Energy-Efficient and Fault-Tolerant Topology Control Game Algorithm for Wireless Sensor Network

**Yongwen Du** * , **Jinzong Xia, Junhui Gong and Xiaohui Hu**

School of Electronics and Information Engineering, Lanzhou Jiaotong University, Lanzhou 730070, China
* Correspondence: duyongwen@mail.lzjtu.cn

**Abstract:** Due to resource constraints and severe conditions, wireless sensor networks should be self-adaptive to maintain certain desirable properties, such as energy efficiency and fault tolerance. In this paper, we design a practical utility function that can effectively balance transmit power, residual energy, and network connectivity, and then we investigate a topology control game model based on non-cooperative game theory. The theoretical analysis shows that the topology game model is a potential game and can converge to a state of the Nash equilibrium. Based on this model, an energy-efficient and fault-tolerant topology control game algorithm, EFTCG, is proposed to adaptively constructs a network topology. In turn, we present two subalgorithms: EFTCG-1 and EFTCG-2. The former just guarantees network single connectivity, but the latter can guarantee network biconnectivity. We evaluate the energy-efficient effect of EFTCG-1. Meanwhile, we also analyze the fault-tolerant performance of EFTCG-2. The simulation results verify the validity of the utility function. EFTCG-1 can efficiently prolong the network lifetime compared with other game-based algorithms, and EFTCG-2 performs better in robustness, although does not significantly reduce the network lifetime.

**Keywords:** wireless sensor network; topology control; game theory; energy efficiency; fault tolerance

## 1. Introduction

Wireless sensor network (WSN) consisting of a large number of tiny sensors with wireless transceivers is a multi-hop and self-organizing network, and it is desirable to many fields such as military, agriculture, medical care, and industry [1]. In WSN, sensors typically operate on batteries and are always randomly deployed in a remote place with harsh or unattended conditions, so it is difficult to recharge or replace batteries. Therefore, energy efficiency is a crucial issue for prolonging network lifetime [2]. What is more, many factors, such as the limitation in resources of nodes and the complex network environment, may lead to failure of active nodes, which will seriously affect system function. For example, if sensor nodes are damaged or their batteries are running out, they cannot communicate with others, which can further lead to network division. Hence, a network also should be fault-tolerant to cope with the emergence of failure nodes [3].

Topology control is an important way to improve performance and prolong the lifetime of WSNs on the premise of maintaining network connectivity by means of adjusting the communication range of wireless sensor nodes [4]. At present, topology control is divided into two essential research directions: power control and hierarchical topology control [5]. Power control is an effective method to get the network topology by adjusting transmit power of nodes, which not only needs to ensure that a network is single connected, biconnected, or multiconnected, but also to avoid hidden terminal and exposed terminal problems as far as possible. Based on clustering idea, hierarchical

topology control constructs the data collection and relay backbone network, which covers all the non-backbone nodes in the network and controls these nodes to sleep at an idle time for hierarchical management. This paper mainly focus on power control of topology control. Many scholars have conducted extensive research this topic. Kirousis [6] simplified power control to a range assignment problem, which is proved to be a NP-hard problem in two and three dimensions. In the work by the authors of [7], the problem of adjusting the transmit powers of each sensor node is formulated a constrained optimization problem with two constraints—single connectivity and biconnectivity—and one optimization objective—maximum power used. Li and Hou [8], based on minimum spanning tree, proposed a topology control algorithm LMST to construct topology using local information; it guarantees that the network is connected and the degree of any node is not greater than 6. In the work by the authors of [9], a fault-tolerant CBTC($\alpha$) algorithm is proposed, in which the generated network is $k$-connected when $\alpha = 2\pi/3k$. These algorithms are traditional topology control algorithms with the precondition that nodes mutually cooperate.

However, in a network, nodes are selfish rather than cooperative, competing with each others to save their resources as much as possible. Game theory is a mathematical method to study the phenomenon of competitive nature [10]. It provides analytical tools to predict the outcome of complex interactions among rational entities. Many scholars have applied a non-cooperative game in game theory to topology control [11,12]. A desired stable solution in non-cooperative game theory is Nash Equilibrium (NE), in which no player has any incentive to unilaterally change his strategy from it [13]. A Nash equilibrium topology is a topology in which no node is interested to unilaterally change its links. Komali [14] formulated the power control as a potential game to guaranteed the existence of a Nash Equilibrium, and proposed a distributed optimal response algorithm MIA. The algorithm may generate different topologies if nodes are executed in a different order. Based on MIA, the authors of [15] proposed an improved algorithm, DIA, which not only guarantees the existence of a Nash Equilibrium, but also ensures the uniqueness of network topology. Nevertheless, the time complexity of DIA is so large that it is not practical in the application of WSN. Zarifzadeh et al. [16] believed that the energy consumption of node is related to its power as well as the traffic load through the node. They provided a practical utility function that contains both of them, then proposed a non-cooperative game-based algorithm, MLPT. However, this algorithm ignores the impact of residual energy on network lifetime. Li et al. [17] proposed a distributed energy balanced topology control algorithm, DEBA, based on potential game. It considered both transmit power and residual energy to balance the energy consumption among nodes. However, the value of influence factors in utility function is not in an order of magnitude so that the impact of them on utility cannot be efficient measured. Moreover, the fixed weights in utility function makes the network lack of adaptation. In the work by te uathors of [18], a self-maintaining topology control game algorithm based on link quality was proposed. It can optimize multiple performances at the same time by synthetically considering communication interference, node degree, residual energy, and link quality in the utility function. However, the weight of this algorithm has a significant influence on practicability, so it is not universal. Although these algorithms have achieved good results in prolonging the network lifetime, they all ignore the important influence of fault tolerance on network function while improving the energy efficiency.

In this paper, we first design a more efficient utility function to balance transmit power, residual energy and network connectivity, then develop a practical model based on the ordinal potential game. The existence of a Nash equilibrium is proved in theoretical analysis. Thus, we propose an energy-efficient and fault-tolerant topology control game algorithm EFTCG. In turn, we presents two subalgorithms. EFTCG-1 can guarantee that network is single connected, and EFTCG-2 maintains network biconnectivity. We evaluate the energy efficiency effect of EFTCG-1. It can balance the energy among nodes and prolong network lifetime in relatively safe environments. Meanwhile, we analyze the fault tolerance of EFTCG-2. It can effectively improve the fault-tolerant ability when network lifetime is appropriately reduced in harsh environments. The simulation results verify the validity of the utility function. It can satisfy different design goals of topology control in different application environments.

The remainder of the paper is organized as follows. Section 2 describes the WSN network model, discusses the problem led by node failure, and introduces the method used in the paper for fault tolerance. Section 3 presents a topology control game model and provides a theoretical analysis. Section 4 proposes an algorithm to get the Nash equilibrium for the game model. Simulation results are presented in Section 5. Finally, we conclude the work in Section 6.

## 2. Network Model and Related Analysis

### 2.1. Network Model

The wireless sensor network can be represented as an undirected graph $G(N, E)$, where $N$ represents a set of sensor nodes in WSN and $E$ is a set of bidirectional links between nodes in set $N$. Also, $N = \{1, 2, ..., n\}$, where $n$ is the number of sensor nodes in WSN and $E \subseteq N^2 = N \times N$.

The transmit power of node determines its communication range. In order to ensure efficient communication, nodes need to adjust their power. $p_{i,j}$ is the minimum power that guarantees the existence of link $(i, j)$ where $p : E \rightarrow \mathbb{R}^+$. $p_i$ is the transmit power of node $i$ and $p_i^{max}$ is the maximum transmit power. The power profile $p = (p_1, p_2, ..., p_n) : p_i = [0, p_i^{max}]$ determines the bidirectional set $E'$ of network. If each node $i$ communicates with maximum power, the formed network is denoted as $G_{max}(N, E_{max})$, where $E_{max} = \{(i, j) | p_i^{max} \geq p_{i,j}, p_j^{max} \geq p_{j,i}\}$. In other words, $E_{max}$ is the set of all possible bidirectional links in the network.

This paper aims to find a subset $E' \subseteq E_{max}$ through the power control method to generate a subgraph, $G'$, that is more efficient than $G_{max}$. The following definitions are given in this paper.

**Definition 1.** *If there is at least one path between any two nodes in an undirected graph G, the graph G is single connected.*

**Definition 2.** *If a single connected component of graph G is divided into two or more connected components after deleting node v and the edges related to v, the node v is called a cut-point.*

**Definition 3.** *If an undirected graph G is connected without cut-points, the graph G is biconnected.*

The network model also needs to satisfy the following assumptions. (1) All nodes are the same type of sensor; they have identical performance parameters, such as wireless transmission characteristic, maximum transmit power, and initial energy, but they have different node ID. (2) Transmit power of node is adjustable, residual energy is different, and battery cannot be recharged except sink node [19]. (3) $G_{max}$ is single connected or biconnected.

### 2.2. Related Analysis

In wireless sensor networks, the reliability of sensor nodes is unsure because of the low-cost and complex application environment [20]. For example, nodes easily fail owing to hardware damage, link instability, and energy exhaustion. Node failure greatly reduces the network's data perception and communication quality, and decreases the stability, reliability, and accuracy of the network. Most WSN topologies only consider the energy-efficient requirements of the network, which aim at efficiently utilizing the limited energy of nodes so as to construct the simplest possible topology structure to prolong the network lifetime. In such a network topology, the probability of node failure is very high, leading to network paralysis. Therefore, only considering energy-efficient topology control can not meet the practical application, so it is important for topology control to consider fault-tolerant.

There are two different types of node failure in WSN: When the sensor node cannot communicate with other nodes when it fails, and when the nodes can communicate normally, but the data they collect or forward is wrong, such as abnormal data generated by external attacks. The node failure involved in this paper are the first type. Failure nodes lose their role in a network. As shown in Figure 1a, the failure nodes A1, A2, and A3 cause nodes A4, A5, and A6 to no longer connect to

other nodes, which divides the network into two disconnected subnetwork, whereas the occurrence of failure nodes B1, B2, and B3 in Figure 1b does not affect the communication between other nodes. It is concluded that different locations of failure nodes have different impacts on network connectivity and availability of other nodes.

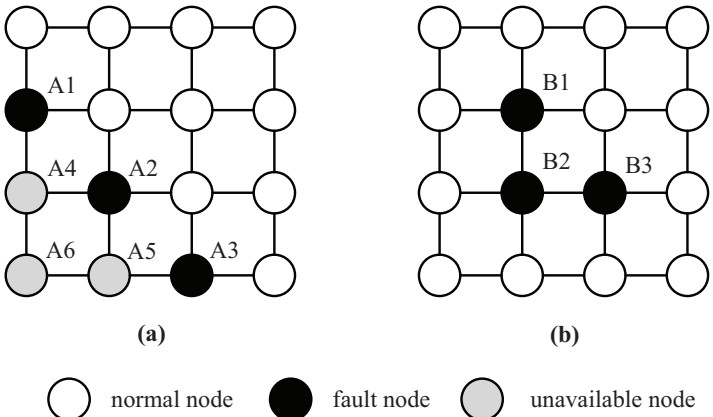

(a)                                          (b)

○ normal node     ● fault node     ◐ unavailable node

**Figure 1.** The effect of failure nodes on a network, (**a**) the failure nodes affect the communication between other nodes; (**b**) the failure nodes do not affect the communication between other nodes.

Therefore, in practical applications, a reasonable topology structure must be fault-tolerant by appropriately retaining some redundancy to deal with possible failure situations. From the perspective of graph theory, a multiconnected graph can improve the network fault tolerance to some extent [21]. Figure 2 shows the network changes after the failure node A1 appears. In Figure 2a, node A2 becomes an isolated node, and the network is divided into two parts. For the biconnected topology in Figure 2b, the network remains connected. It can be seen by analysis that a biconnected network can eliminate the different impact on topology caused by the location of single failure node.

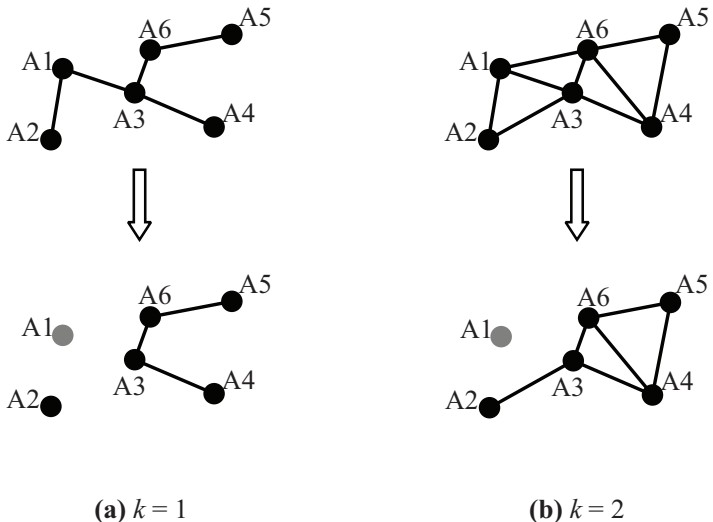

(**a**) $k = 1$                           (**b**) $k = 2$

**Figure 2.** The effect of connectivity $k$ on a network, (**a**) the change of a single connectivity network after single failure node appears; (**b**) the change of a biconnectivity network after single failure node appears.

Generally, the higher the connectivity $k$ of a network is, the greater the number of failure nodes that network can simultaneously tolerate [22]. However, with the increase of connectivity, its cost will also rise sharply [23]. In this paper, we just maintain network biconnectivity to study the validity of the utility function we designed in the analysis of fault-tolerant experiments.

The identification of cut-points in a network is the premise of judging the network biconnectivity. Depth-first search (DFS) is used to determine whether a node is a cut-point. A DFS spanning tree

can be obtained in DFS process. Figure 3a is an undirected graph, and Figure 3b is its DFS spanning tree that DFS starts from node A. In the depth-first search process, the starting node is called a root node; the edge that passes through the unvisited node is a tree edge, such as the solid line in Figure 3b; and the edge that passes through the visited node is a back edge, such as the dotted line in Figure 3b.

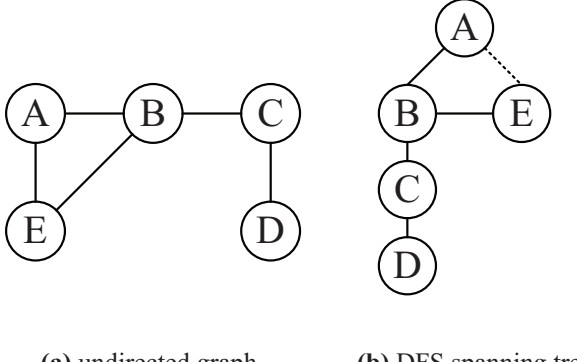

**(a)** undirected graph       **(b)** DFS spanning tree

**Figure 3.** Depth-first search (DFS) spanning tree of an undirected graph, (**a**) an undirected graph; (**b**) a DFS spanning tree accessed from node A.

The characteristic of cut-point can be obtained from the DFS spanning tree. There are two kinds of cut-points [24]:

- If a root node $t$ has two or more subtrees [25], then node $t$ is a cut-point.
- If a node that in the subtree of non-leaf node $v$ does not have a back edge pointing to the ancestor of node $v$, then node $v$ is a cut-point.

For a root node $t$, we can easily judge whether it is a cut-point according to the number of its subtrees. If $t$ has at least two subtrees, its subtrees can not visit each other after deleting $t$. For non-leaf node $v$, the judgment is a little complicated. In the depth-first search process, we use time($v$) to represent a timestamp when node $v$ is first traversed and low($v$) to represent a timestamp of the earliest node that node $v$ can access without passing through its parent node. The formula of low($v$) is as follows,

$$\text{low}(v) = \begin{cases} \min\{\text{low}(v), \text{low}(w)\} & \text{(I)} \\ \min\{\text{low}(v), \text{time}(w)\} & \text{(II)} \end{cases}$$

where condition I states that edge $(v, w)$ is tree edge and condition II states that edge $(v, w)$ is the back edge and node $w$ is not parent of node $v$. low($v$) $\geq$ time($w$) indicates that node $w$ and its descendant nodes do not point back to ancestor of node $v$. In this case, after deleting node $v$, the graph has more than one disconnected components, so obviously node $v$ is a cut-point.

## 3. Topology Control Model Based on Game Theory

### 3.1. Game Theory

Due to the nodes are selfish and all nodes want to maximize their own benefit, we introduce the non-cooperative game model that address the interaction among individual rational decision-makers into topology control [26]. In the most straightforward approach, the rational decision-makers are referred to as "players", and players select a single strategy from a set of feasible strategies. Interaction between the players is represented by the influence that each player has on the resulting outcome after all players have selected their actions. Each player evaluates the resulting outcome through utility function representing her objectives [27]. There are two ways of representing different components (players, strategies, and utility function) of a game: normal, or strategic form, and extensive. Here, we will focus on the normal form representation. Formally, a normal form of a game is given by $\Gamma\langle N, S, \{u_i\}\rangle$, where consists of three elements [28]:

- Player set $N$. The set of players in the game can be expressed as $N = \{1, 2, ..., n\}$, where $n$ is the number of players.
- Strategy space $S$. The set of strategies for player $i$ is denoted as $S_i$. If $i$ has $m$ alternative strategies, $p$, then $S_i = \{s_i^1, s_i^2, ..., s_i^m\}$, which is abbreviated as $S_i = \{s_1, s_2, ..., s_m\}$, so strategy space $S = \times_{i=1}^n S_i$. $s = (s_i, s_{-i})$ denotes a strategy profile, where $s_i$ is the strategy of player $i$ and $s_{-i}$ means the strategies of the other $n - 1$ players.
- Utility function $u_i$ of player $i$. $u_i : S \to \mathbb{R}$ represents the utility of the $i$-th player under the strategy combination $(s_i, s_{-i})$. $\{u_i\} = \{u_1, u_2, ..., u_n\} : S \to \mathbb{R}^n$ denotes the set of such utility functions.

An important concept in non-cooperative game theory is Nash equilibrium (NE) [29]. This solution concept is defined as a stable point because no player has any incentive to unilaterally change his or her action from it. When the game achieves the NE, if any node changes its strategy alone, the income of this node will suffer a loss, that is, at this equilibrium state, all players do not want to take the initiative to deviate from this state. The definition of Nash equilibrium is given as follows.

**Definition 4.** *For $\forall i \in N$ and $\forall s_i \in S_i$, a strategy combination $s^* = (s_i^*, s_{-i}^*)$ is a Nash Equilibrium of game $\Gamma \langle N, S, \{u_i\} \rangle$, if we have*

$$u_i(s^*) \geq u_i(s_i, s_{-i}^*). \tag{1}$$

The existence of a Nash equilibrium is vital to the combinatorial optimization problem in game theory. A game may possess a large number of NEs or none at all. Some classes of games are known to possess at least one NE. One such class is the potential games with compact action spaces, which are known to possess at least one NE in pure strategies [30]. A potential game is a normal form game such that any change in the utility function of any player due to a unilateral deviation by that player is correspondingly reflected in a global function, referred to as the potential function. Ordinal potential game is a potential game [31], its definition is shown as follows.

**Definition 5.** *A strategy game $\Gamma \langle N, S, \{u_i\} \rangle$ is an ordinal potential game if there is a function $V : S \to \mathbb{R}$ to satisfy*

$$V(p_i, s_{-i}) - V(q_i, s_{-i}) > 0 \Leftrightarrow u_i(p_i, s_{-i}) - u_i(q_i, s_{-i}) > 0, \tag{2}$$

*where $\forall i \in N$, $\forall s_{-i} \in S_i$ and $\forall p_i, q_i \in S_i$. The function $V$ is called an ordinal potential function of the strategy game $\Gamma$.*

**Theorem 1.** *If the strategy game $\Gamma \langle N, S, \{u_i\} \rangle$ is an ordinal potential game and $V$ is its ordinal potential function, then the strategy combination $s^*$ of maximizing $V$ is a Nash equilibrium of the game $\Gamma$.*

*3.2. Topology Control Game Model*

In this section, we establish the topology control game model $T \langle N, S, \{u_i\} \rangle$, where $N$ is the set of all sensor nodes in a WSN and the strategy set $S_i$ of node $i$ is its optional power set. In the establishment of the topology control game theory, the most important thing is to determine the utility function.

The utility function maps the transmit power of a node to a utility value of the node, and the utility value of a node refers to the value of a benefit that the node gets as a result of its strategy. However, it is difficult to assess how a node will value different levels of performance and what trade-offs it is willing to make, so we design the utility function that meets some needs from the following aspects:

- Transmit power.

  Energy efficiency is one of the most crucial requirements in WSNs. Hence, it is imperative for every node to be energy efficient: this not only increases the node's own operational lifetime but also contributes to an overall increase in the network lifetime. Thus energy is a limiting factor for network operability. Transmitting with low power is one way of conserving energy. So the transmit power must be considered in the utility function.

- Residual energy.

  Energy consumption balance is also very important for maximizing network lifetime. When the residual energy consumption of nodes in the network is uneven, some nodes will die prematurely to affect normal network work. So the residual energy of a node should also be considered in the utility function. In this paper, residual energy and initial energy of node $i$ are expressed as $E_r(i)$ and $E_o(i)$, respectively.

- Network connectivity

  The premise of topology control is to preserve network connectivity. While nodes change network topology by adjusting their transmit power, it is necessary to ensure that the network is still $k$-connected ($k$ = 1, 2). $k$-connected function is denoted as $f_k(p_i, p_{-i})$, where $p_i$ represents the transmit power of node $i$ and $p_{-i}$ represents the transmit powers of other nodes. $f_k(p_i, p_{-i}) = 1$ denotes that network is $k$-connected and otherwise network is not $k$-connected. Also $\forall i \in N$ and $p_i > q_i$, $f_k(p_i, p_{-i}) \geq f_k(q_i, p_{-i})$. Obviously, the function $f_k$ is monotonic and nondecreasing.

Therefore, the utility function $u_i$ is defined as following, which describes the trade-off of the above various influence factors.

$$u_i(p_i, p_{-i}) = f_k(p_i, p_{-i}) \left( \alpha_i \frac{p_i^{\max} - p_i}{p_i^{\max}} + \beta_i \overline{E_i(p_i)} \right) \tag{3}$$

Here, $\alpha_i$ and $\beta_i$ are the weights and both are positive numbers; $\overline{E_i(p_i)} = \frac{1}{m} \sum_{j=1}^{m} \frac{E_r(j)}{E_o(j)}$, which characterize the average residual energy level of neighbor nodes of node $i$, where $j$ is the node that node $i$ can reach by one-hop with power $p_i$ and $m$ is the number of neighbor nodes.

In the utility function, $f_k(p_i, p_{-i})$ ensures that the network is still $k$-connected after multiple rounds of a game when node increase utility by adjusting power, and $\alpha_i \frac{(p_i^{\max} - p_i)}{p_i^{\max}} + \beta_i \overline{E_i(p_i)}$ represents the benefit when network is $k$-connected. Network lifetime is closely related to the lifetime of node, and the main factors affecting node lifetime are the transmit power and the residual energy. When other factors remain unchanged, the transmit power is negatively correlated with node lifetime, while the residual energy is positively correlated. On the one hand, the node will reduce its transmit power to save energy. On the other hand, it will tend to choose neighbor nodes with more residual energy to forward data. If the average residual energy of its neighbor nodes is low, they may die prematurely, thus affecting node $i$ forwarding data and even causing a network to stop working.

Because the energy is not of the same order of magnitude with the transmit power, the impact of them on node lifetime can't be effectively compared. So we first normalize them and then gives them corresponding weights. The weights [17,18] of other topology control algorithms based on game theory are fixed. However, each node in a network is an independent individual, it will make decisions according to its own situation. Thus, the weights of the utility function should be self-adaptively adjusted instead of fixed value. In this paper, $\alpha_i$ and $\beta_i$ are determined by its residual energy where $\alpha_i = 1 - \frac{E_r(i)}{E_o(i)}$ and $\beta_i = 1 - \alpha_i$. If the residual energy of node $i$ become smaller, $\alpha_i$ will become larger and $\beta_i$ will become smaller, which makes node $i$ more inclined to reduce its power to prolong its lifetime. For example, when the energy of the node is too low, $\alpha_i$ far greater than $\beta_i$ it will first reduce its transmit power to reduce its energy consumption, rather than increasing the power to improve the average residual energy of its neighbor nodes.

### 3.3. Model Analysis

We begin by defining a global function in Equation (4) based on the game $T\langle N, S, \{u_i\} \rangle$ with the objective function of each node given by Equation (3). Then, we prove that the function is an ordinal potential function, so the game $T\langle N, S, \{u_i\} \rangle$ is an ordinal potential game in Theorem 2. According to the convergence property of ordinal potential game, the Nash equilibrium is guaranteed in our game model in Theorem 3.

**Theorem 2.** *The topology control game model* $T\langle N, S, \{u_i\}\rangle$ *is an ordinal potential game. Its ordinal potential function is defined as*

$$V(p_i, p_{-i}) = \sum_{i \in N} \left[ f_k^i(p_i, p_{-i}) \left( \alpha_i \frac{p_i^{\max} - p_i}{p_i^{\max}} + \beta_i \overline{E_i(p_i)} \right) \right] \tag{4}$$

**Proof.** We prove the preceding theorem by applying the definition of the ordinal potential function. First, if node $i$ unilaterally changes its strategy form $p_i$ to $q_i$, the difference of utility according to the function in Equation (3) can be described as follows.

$$\begin{aligned}
\Delta u_i &= u_i(p_i, p_{-i}) - u_i(q_i, p_{-i}) \\
&= f_k(p_i, p_{-i}) \left( \alpha_i \frac{p_i^{\max} - p_i}{p_i^{\max}} + \beta_i \overline{E_i(p_i)} \right) \\
&\quad - f_k(q_i, p_{-i}) \left( \alpha_i \frac{p_i^{\max} - q_i}{p_i^{\max}} + \beta_i \overline{E_i(q_i)} \right) \\
&= \frac{\alpha_i}{p_i^{\max}} \left[ f_k(p_i, p_{-i})(p_i^{\max} - p_i) - f_k(q_i, p_{-i})(p_i^{\max} - q_i) \right] \\
&\quad + \beta_i \left[ f_k(p_i, p_{-i}) \overline{E_i(p_i)} - [f_k(q_i, p_{-i}) \overline{E_i(q_i)} \right].
\end{aligned} \tag{5}$$

Similarly, the difference of ordinal potential functions in Equation (4) is as follows.

$$\begin{aligned}
\Delta V &= V(p_i, p_{-i}) - V(q_i, p_{-i}) \\
&= \sum_{i \in N} \left[ f_k^i(p_i, p_{-i}) \left( \alpha_i \frac{p_i^{\max} - p_i}{p_i^{\max}} + \beta_i \overline{E_i(p_i)} \right) \right] \\
&\quad - \sum_{i \in N} \left[ f_k^i(q_i, p_{-i}) \left( \alpha_i \frac{p_i^{\max} - q_i}{p_i^{\max}} + \beta_i \overline{E_i(q_i)} \right) \right] \\
&= \frac{\alpha_i}{p_i^{\max}} \left[ f_k^i(p_i, p_{-i})(p_i^{\max} - p_i) - f_k^i(q_i, p_{-i})(p_i^{\max} - q_i) \right] \\
&\quad + \beta_i \left[ f_k^i(p_i, p_{-i}) \overline{E_i(p_i)} - [f_k^i(q_i, p_{-i}) \overline{E_i(q_i)} \right] \\
&\quad + \sum_{j \in N, j \neq i} \left[ f_k^j(p_i, p_{-i}) - f_k^j(q_i, p_{-i}) \right] \left( \alpha_j \frac{p_j^{\max} - p_j}{p_j^{\max}} + \beta_i \overline{E_j(p_j)} \right)
\end{aligned} \tag{6}$$

We let $\lambda = \sum_{j \in N, j \neq i} \left[ f_k^j(p_i, p_{-i}) - f_k^j(q_i, p_{-i}) \right] \left( \alpha_j \frac{p_j^{\max} - p_j}{p_j^{\max}} + \beta_i \overline{E_j(p_j)} \right)$, so it follows from Equations (5) and (6) that

$$\Delta V = \begin{cases}
\Delta u_i + \lambda & if \quad p_i > q_i \quad and \quad f_k^i(p_i) > f_k^i(q_i) \\
\Delta u_i & if \quad p_i > q_i \quad and \quad f_k^i(p_i) = f_k^i(q_i) = 1 \\
0 & if \quad p_i > q_i \quad and \quad f_k^i(p_i) = f_k^i(q_i) = 0 \\
\Delta u_i + \lambda & if \quad p_i < q_i \quad and \quad f_k^i(p_i) < f_k^i(q_i) \\
\Delta u_i & if \quad p_i < q_i \quad and \quad f_k^i(p_i) = f_k^i(q_i) = 1 \\
0 & if \quad p_i < q_i \quad and \quad f_k^i(p_i) = f_k^i(q_i) = 0
\end{cases} \tag{7}$$

If $p_i > q_i$ and $f_k^i(p_i) > f_k^i(q_i)$, that is $f_k^i(p_i) = 1$ and $f_k^i(q_i) = 0$, so $\Delta u_i > 0$ and $\lambda > 0$. If $p_i < q_i$ and $f_k^i(p_i) > f_k^i(q_i)$, that is, $f_k^i(p_i) = 0$ and $f_k^i(q_i) = 1$, so $\Delta u_i < 0$ and $\lambda < 0$. In summary, we can see that the signs of $\Delta V$ and $\Delta u_i$ are consistent. Therefore, the game $T\langle N, S, \{u_i\}\rangle$ is an ordinary potential game and $V(p_i, p_{-i})$ is its ordinary potential function according to Definition 5. $\square$

**Theorem 3.** *There must be a Nash equilibrium in game* $T\langle N, S, \{u_i\}\rangle$.

**Proof.** Theorem 1 and 2 show that the game $T\langle N, S, \{u_i\}\rangle$ is an ordinal potential game and the strategy combination of maximizing the ordinal potential function $V$ is a Nash Equilibrium. Because the optional strategy of node $i$ is limited, there must exist a strategy combination to maximize the $V$, and the combination of strategy is a Nash Equilibrium of the game. $\square$

## 4. Topology Control Game Algorithm

In this section, we propose an energy-efficient and fault-tolerant topology control game (EFTCG) algorithm. Although Theorem 3 proves that the game model can converge to NE, it not guaranteed that the game algorithm can also converge to the NE state. Therefore, in order to ensure that the participants can eventual converge to the NE state, the better response strategy convergence mechanism is used. The better response strategy is defined as follows.

**Definition 6.** *In any round of a game, once a player i is allowed to update its strategy, it chooses a strategy that has a higher utility than the current strategy. More concisely, the current round of a game is $r$, $r = 0, 1, ..., m - 1$. Let $s_{i,r}$ be the current strategy of node i under the current round, given the strategy of other players, each player chooses a strategy next by*

$$s_{i,r+1} = \underset{s_i \in \{s_{i,r}, s_i^{(r+1)}\}}{\arg\max} \ u_i(s_i, s_{-i}) \tag{8}$$

The algorithm designed in this paper mainly includes two phases: Section 4.1, Topology Information Collection Phase, and Section 4.2, Topology Game Phase.

### 4.1. Topology Information Collection Phase

Each sensor node in a network needs to collect certain topology-related information when making topology decision. The main task of the phase is to collect these decision pieces of information.

Firstly, each node $i$ initializes its transmit power with the maximum power $p_i^{\max}$. Then, it discovers its neighbors by broadcasting "Hello Message", including its node ID and maximum transmit power, and collects the responses provided by each receiver.

Upon successful reception of ACK message from each responding neighbor node $j$ at power $p_j^{\max}$, which contains its node ID, residual energy and maximum power, node $i$ adds neighbor $j$'s information and $p_{i,j}$ into its neighbor list where the minimum transmit $p_{i,j}$ required to establish link $(i,j)$ can be determined in the work by the authors of [32].

### 4.2. Topology Game Phase

On the basis of the collected neighbor lists in the collection phase, the strategy set $S_i = \{p_{\max} = p_1, p_2, ..., p_m = p_{\min}\}$ of node $i$ is formed in descending order, where $m$ represents the number of neighbors.

In any round of the game, all the nodes execute sequentially the game for adjusting power according to node ID, and only one node is allowed to update its strategy at the same time [33]. In order to converge to a state of Nash equilibrium in finite steps, the better response strategy [17] is adopted. Node $i$ chooses transmit power next according to Equation (8). In other words, node $i$ chooses the power if selected strategy gives a better utility than its current strategy, and broadcasts the new power strategy to other nodes. Otherwise, the node will keep the current power unchanged. With the continuation of the game, the strategy of each node will eventually converge to Nash equilibrium. The procedure of this phase is shown in Algorithm 1.

---

**Algorithm 1** Topology Game Phase Algorithm

---

**Input:** Nodes' neighbor lists
**Output:** Nodes' transmit power sets

 1: $r = 0$
 2: $p_i^* = p_i^{\max}, \forall i \in N$
 3: $s = \{p_1^*, p_2^*, ..., p_n^*\}$
 4: **while** $s$ is not an $NE$ **do**

 5:  $\quad r = r + 1$
 6:  $\quad$ **for** Node $i$ in $N$ **do**

 7:  $\qquad$ Node $i$ choose a power according to $p_i^* = \underset{q_i \in \{p_i^{(r)}, p_i^*\}}{\arg\max}\ u_i(q_i, p_{-i})$

 8:  $\qquad$ **if** $p_i^*$ has changed **then**
 9:  $\qquad\quad$ Broadcast a Hello Message including the new power $p_i^*$ at $p_i^{\max}$
10:  $\qquad$ **end if**
11:  $\quad$ **end for**
12:  $\quad$ **if** $s$ has not changed **then**

13:  $\qquad$ $s$ is an $NE$
14:  $\quad$ **end if**
15: **end while**
16: **return** $s$

---

*4.3. Game Analysis*

**Theorem 4.** *The complexity of EFTCG algorithm is O(n), where n is the number of nodes in the network.*

**Proof.** Suppose that the number of nodes in the network is $n$ and the number of neighbors of one node is $m$ ($m$ is much less than $n$). In the Information Collection Phase of the algorithm, $n$ nodes need to initialize the maximum power $p_{\max}$ to broadcast data packets containing their own information. When any node receives the broadcast information, it sends the response message. After each node receives the response information of the reachable node, it determines its own neighbor list. Then any node gets the optional transmit power strategy set and broadcasts it. So the number of information collection in this phase is $3n$. Obviously, the total number of information exchanges is a function of $3n$. In the Game Phase, we adopt a better response strategy approach: each node has two transmit powers to play a game in turn. After each node determines its transmit power, it sends a packet with adjusted transmit power to inform other nodes to update their neighbor lists, so the complexity of one phase is $2n + n$. At this phase, the algorithm will converge after $m$ rounds at most. Then the complexity of this phase is: $m(2n + n)$. Therefore, the complexity of EFTCG algorithm is $O(n)$. $\quad\square$

**5. Simulation Results**

In this section, we evaluate the performance of EFTCG via simulations and compare it with other game-based algorithms. The simulations were conducted using Python. We assume that sensor nodes are randomly distributed and immovable, and only one sink node locates in the center of the monitoring area. The simulation parameters are shown in Table 1. Source code for the algorithm is available at https://github.com/duyongwen/EFTCG.

**Table 1.** Experimental parameters.

| Parameters | Value |
| --- | --- |
| Monitoring area | 300 m × 300 m |
| The maximum transmit distance | 100 m |
| Initial energy | 50 J |
| Residual energy | Poisson distribution with parameter of 25 |
| The wavelength | 0.1224 m |
| Receiving threshold | $7 \times 10^{-10}$ w |
| Transmit antenna gain | 1 |
| Receiving antenna gain | 1 |
| System loss | 1 |

*5.1. Analysis of Energy-Efficiency*

We use EFTCG to refer to EFTCG-1 without causing confusion in this subsection. Its energy-efficient effect is evaluated by comparison with DIA [15], MLPT [16], and DEBA [17]. We compare them in terms of the standard deviation of the residual energy, the average transmit power, the average node degree, the average hop, and the network time.

To make an intuitive comparison of topology structures formed by the four algorithms, 100 nodes are randomly distributed in the monitoring area. The network topology constructed by four algorithms is shown in Figure 4, where the number is the residual energy of a node. It can be seen from the figure, that in topologies (a) and (b), constructed by the DIA algorithm and the MLPT algorithm, there are a relatively large number of the bottleneck nodes with less residual energy and heavier load than surrounding neighbors. These nodes are likely to die prematurely due to the less residual energy but the large load when the residual energy of other nodes are high, which affects the connectivity of the network. Compared with the DIA and MLPT algorithms, the topology constructed by the DEBA algorithm or EFTCG algorithm can effectively reduce the bottleneck nodes generated by the first two algorithms, and make the nodes with more residual energy participate in the data forwarding of the network. In addition, the average node degree of topology (c) is intuitively higher than of topology (d), which greatly increases the energy consumption of nodes in topology (c).

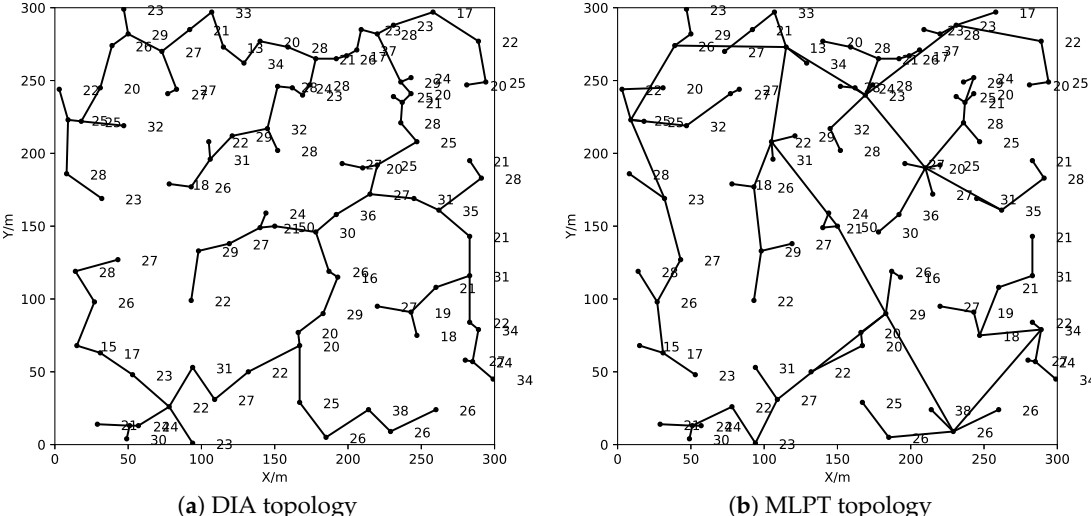

(**a**) DIA topology      (**b**) MLPT topology

**Figure 4.** *Cont.*

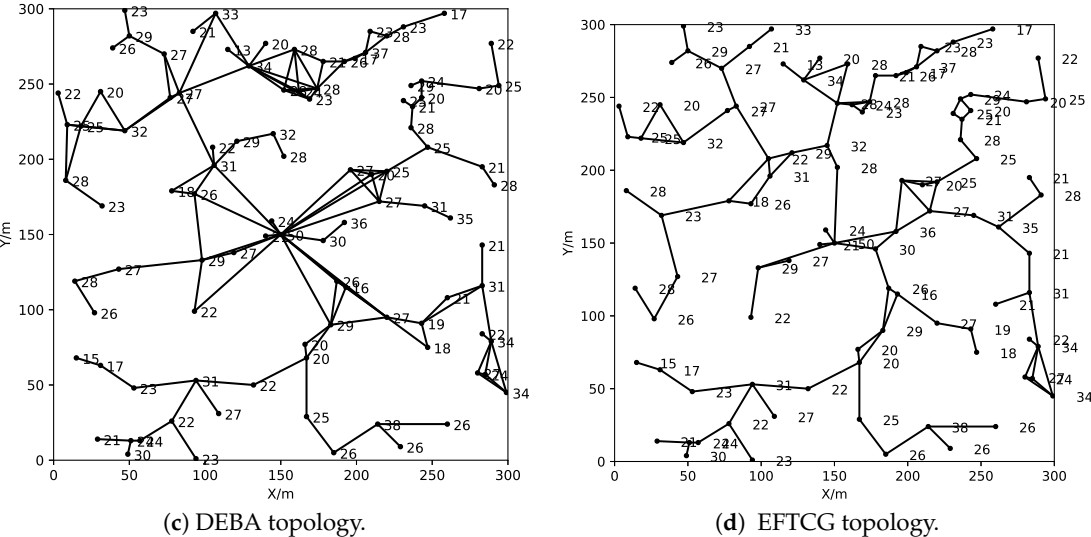

**(c)** DEBA topology.

**(d)** EFTCG topology.

**Figure 4.** Topologies derived under different algorithms, (**a**) a topology derived under DIA; (**b**) a topology derived under MLPT; (**c**) a topology derived under DEBA; (**d**) a topology derived under EFTCG.

Figure 5 shows the comparison of the standard deviation of the residual energy of nodes under topologies in Figure 4. Over time, the residual energy of nodes becomes more and more unbalanced. The residual energy standard deviation of EFTCG is lowest than the others, and the rising speed is also slow. DIA and MLPT do not consider energy, which makes the energy consumption unbalanced. Although DEBA pays attention to energy, it increases the average transmit power of nodes. Therefore, the residual energy standard deviation of nodes rises fast. There is a better trade-offs between transmit power and residual energy in EFTCG, so that energy of nodes is more balanced when the network runs.

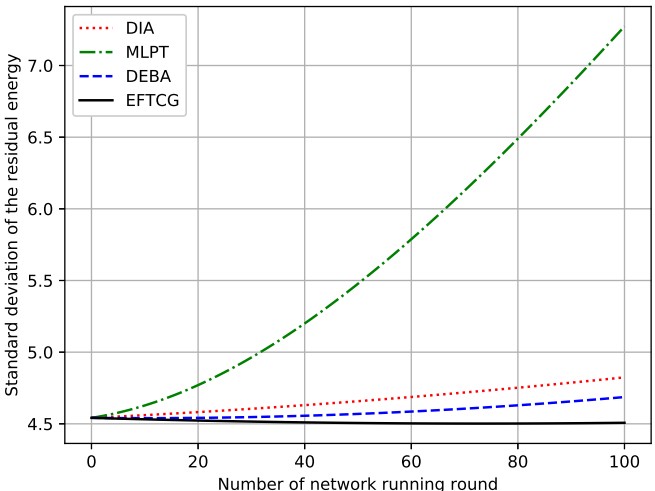

**Figure 5.** The comparison of standard deviation of the residual energy under different algorithms.

In the next simulation, we vary the number of nodes in the region from 30 to 100 to change the node density. For each node density, simulations are executed ten times, and resulting values are obtained by calculating the averaging of these ten simulations.

From the Figure 6, the average transmit power decrease with the increase of the number of nodes. Compared with MLPT and DEBA algorithms, EFTCG algorithm has lower average transmit power, but EFTCG consumes slightly more power than DIA. DIA only aims at minimizing power, so nodes can get the lowest transmit power. Compared with the DEBA algorithm, the EFTCG algorithm has better performance in reducing network energy consumption by reducing transmit power.

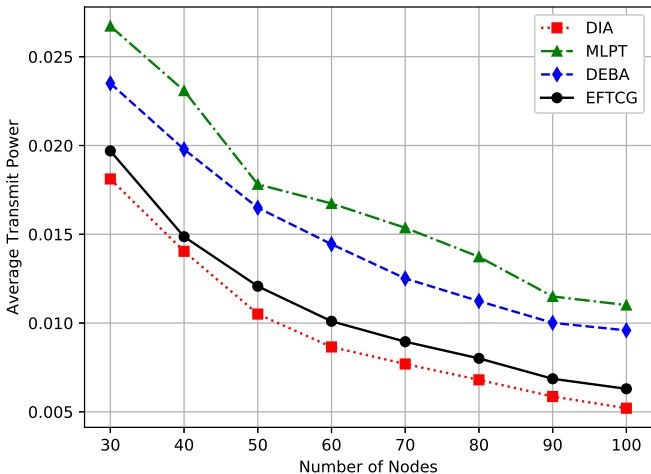

**Figure 6.** The comparison of the average transmit power under different algorithms.

As shown in Figure 7, the average node degree becomes large as the node density increases. The DEBA algorithm tends to make nodes communicate with other nodes with more residual energy, so it has the highest degree. In the EFTCG algorithm, when a node has much energy, it may increase its transmit power within the affordable range to select the neighbor nodes with more residual energy to forward data, so the average degree is higher than DIA and DEBA. The degree of EFTCG is relatively moderate, which not only does not consume too much energy but also can forward data efficiently.

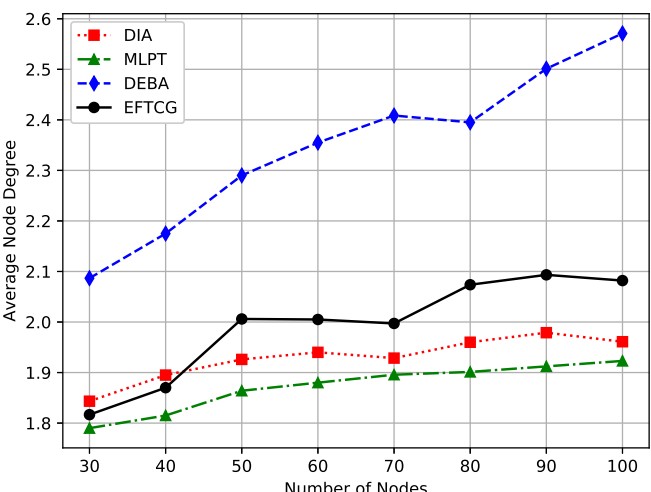

**Figure 7.** The comparison of the average node degree under different algorithms.

Figure 8 shows a comparison of the average-hop under the shortest path between an ordinary node and a sink node. As the number of node increase, the average-hop increases. The average-hop of EFTCG is slightly higher than of DEBA and MLPT, but lower than DIA. Under the topologies of DEBA and MLPT, the average transmit power of nodes are higher, the communication radius of their nodes is also relatively large. So the number of nodes passing through when reaching the sink node is relatively small at the same distance. Since the node of EFTCG operates at lower power, it is inevitable to increase its hop count from source to destination.

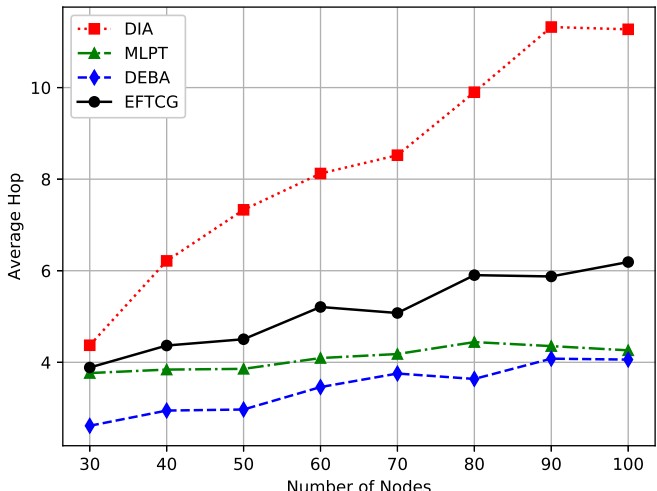

**Figure 8.** The comparison of the average hop under different algorithms

Figure 9 shows the comparison of network lifetime. In order to objectively compare the lifetime of different algorithms, this paper uses the number of networks running round when the first node dies as the network lifetime [32]. As the number of nodes increases, the network lifetime decreases gradually. The network lifetime of EFTCG and DIA is better than the other two algorithms, and there is little difference between them. EFTCG effectively balances power and energy, so it has low power and moderate node size, which significantly prolongs the network lifetime. The complexity of DIA is $O(n^2)$ [15] and EFTCG is $O(n)$. The higher complexity will incur much more overhead [34], so EFTCG has higher efficiency in network lifetime than DIA. EFTCG effectively prolongs the network lifetime and has lower complexity, which is of considerable significance to the deployment of wireless sensor networks.

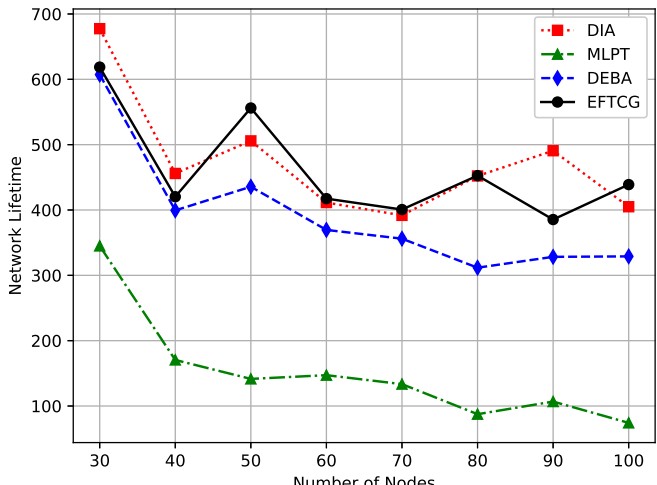

**Figure 9.** The comparison of the network lifetime under different algorithms.

### 5.2. Analysis of Fault-Tolerant

In this section, fault-tolerant analysis is carried out for the EFTCG-1, EFTCG-2, and DEBA algorithms. The fault nodes are randomly generated in sequence, and the maximum number of fault nodes is 10 in this simulation experiment.

The fault tolerance of the network is used to measure the ability of the remaining nodes to maintain connectivity when some nodes fail. In this paper, the degree of fault tolerance is measured by the rate of survival nodes, the rate of connectable node pairs, and average node degree. The following is the definition of the fault-tolerant index used in this paper.

**Definition 7.** *The rate of survival nodes, $R_s$, is calculated according to the following formula.*

$$R_s = \frac{n - n_f - n_u}{n} \times 100\%, \tag{9}$$

*where n represents the total number of nodes in the network, $n_f$ represents the numbers of failure nodes, $n_u$ represents the number of unavailable nodes, and $n - n_f - n_u$ is the number of survival nodes.*

**Definition 8.** *The rate of connectable node pairs $R_{cnp}$ is defined as follows.*

$$R_{cnp} = \frac{1}{n(n-1)} \sum_{i \in N_s} \sum_{j > i} l_{ij} \times 100\%, \tag{10}$$

*where $N_s$ is the set of survival nodes in network and $\sum_{i \in N_r} \sum_{j > i} l_{ij}$ represents the total numbers of connectable node pairs. If there is communication link between i and j then $l_{ij} = 1$; otherwise, $l_{ij} = 0$.*

**Definition 9.** *The node degree can reflect the robustness of a network in some sense. It usually refers to the number of edges directly connected with neighbors. The calculation formula of degree $d_i$ of node i is*

$$d_i = \sum_{j \in N, j \neq i} l_{ij}, \tag{11}$$

*where if node i and j are directly connected, then $l_{ij} = 1$, otherwise $l_{ij} = 0$.*

*The average node degree D of network is the average degree of all nodes. The calculate formula is*

$$D = \frac{\sum_{i=1}^{n} d_i}{n} \tag{12}$$

In order to visually compare the topology formed by the EFTCG-1 and EFTCG-2 algorithms, 80 nodes are randomly distributed in the monitoring area. As shown in Figure 10, the number of links in the topology constructed by the EFTCG-2 algorithm is significantly increased, because it adds redundant links to ensure network biconnectivity.

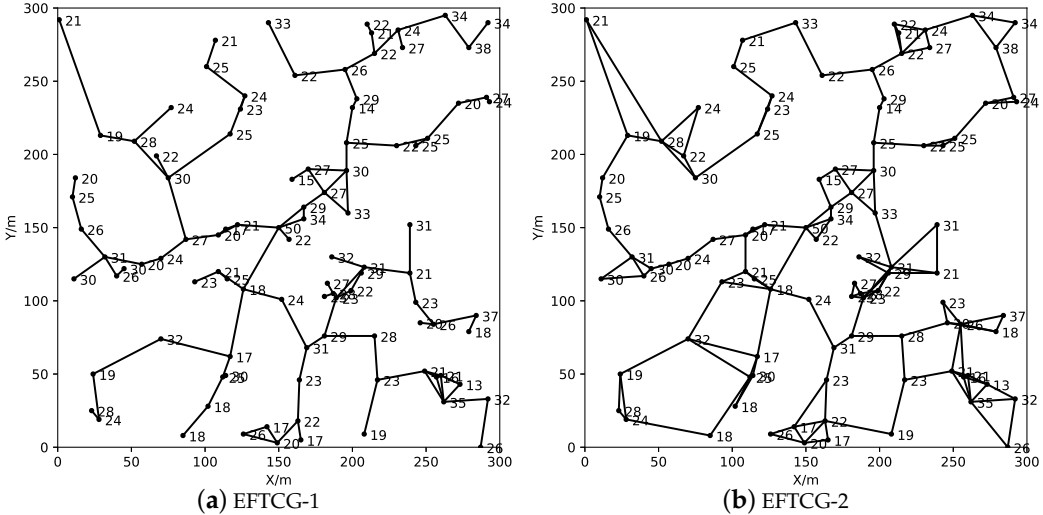

(**a**) EFTCG-1      (**b**) EFTCG-2

**Figure 10.** Topologies under different connectivity, (**a**) a topology derived under EFTCG-1; (**b**) a topology derived under EFTCG-2.

In the next simulation, we vary the number of nodes from 40 to 100 to change the node density. For each node density, simulations are executed ten times, and resulting values are obtained by calculating the averaging of these ten simulations.

In Figure 11a, when the number of nodes is small, the average transmit power of EFTCG-2 is higher than EFTCG-1 and DEBA, but as the number of nodes increases, the average transmit power is equal to the DEBA. EFTCG-2 makes the node properly increase its transmit power to add links to maintain the network biconnectivity, so that its average transmit power is the highest, which makes that its average node degree is also larger than others in Figure 11b.

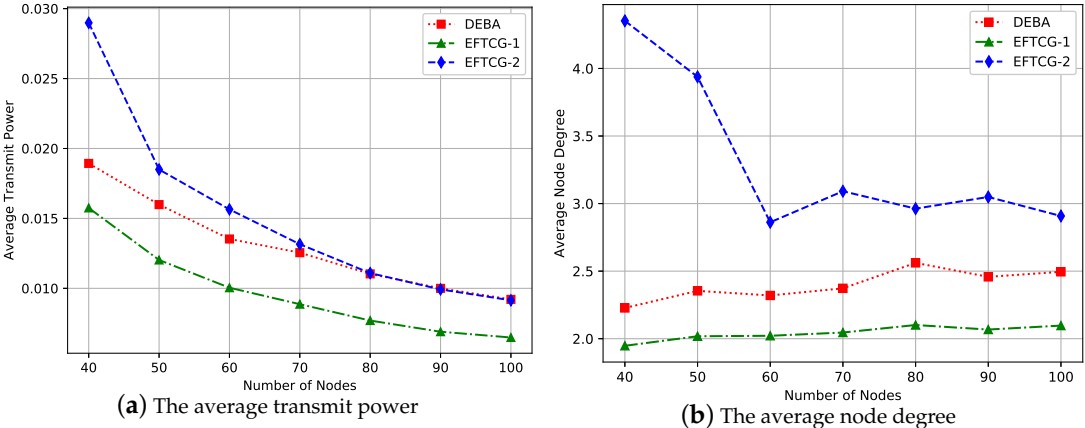

**Figure 11.** The average transmit power and the average node degree, (**a**) the change of the average transmit power; (**b**) the change of the average node degree.

Figure 12a shows that the survival nodes decrease with the increase of the number of failure nodes in one experiment simulation, and the EFTCG-2 algorithm decreases more smoothly, whereas the other two algorithms fluctuate more. Figure 12b indicates that the variation of the rate of survival nodes under a different number of nodes after ten failure nodes appear in the network, and EFTCG-2 has more survival nodes than EFTCG-1 and DEBA. As shown in Figure 13a, with the number of fault nodes increases, the number of connectable node pairs decreases gradually, which is consistent with the trend in Figure 12. Similarly, Figure 13b indicates a variation after ten failure nodes appear in the network, and EFTCG-2 has more links than the other two algorithms. The node that acts as a cut-point in the network model are called "separated nodes". Once a separated node fails, it will cause more nodes to be unavailable, resulting in a sharp decrease. EFTCG-2 can effectively decrease the number of separated nodes by adding redundant links to reduce the number of nodes that are unavailable when node failure occurs, improving the fault tolerance of the network.

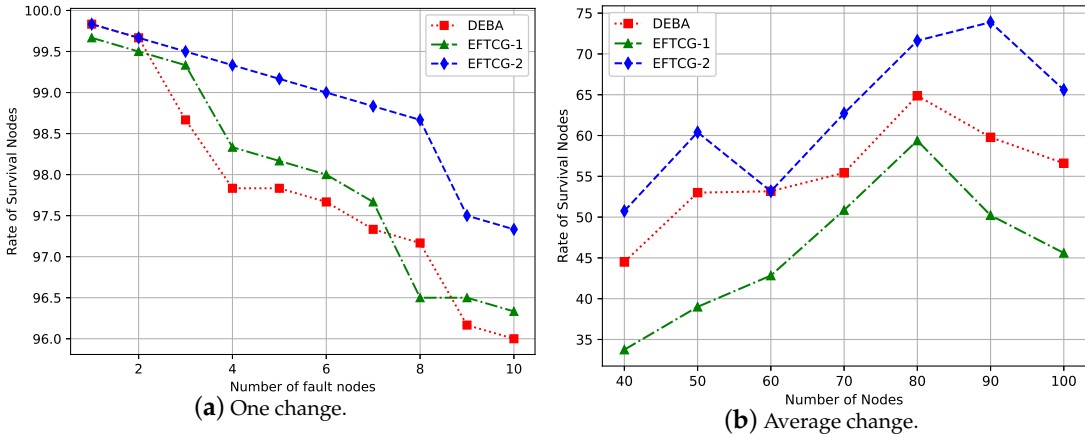

**Figure 12.** The rate of connectable node pairs, (**a**) the change of the rate of connectable node pairs with the generation of failure nodes when the number of nodes is 100; (**b**) the change of the rate of connectable node pairs under different node density after 10 failure nodes appear.

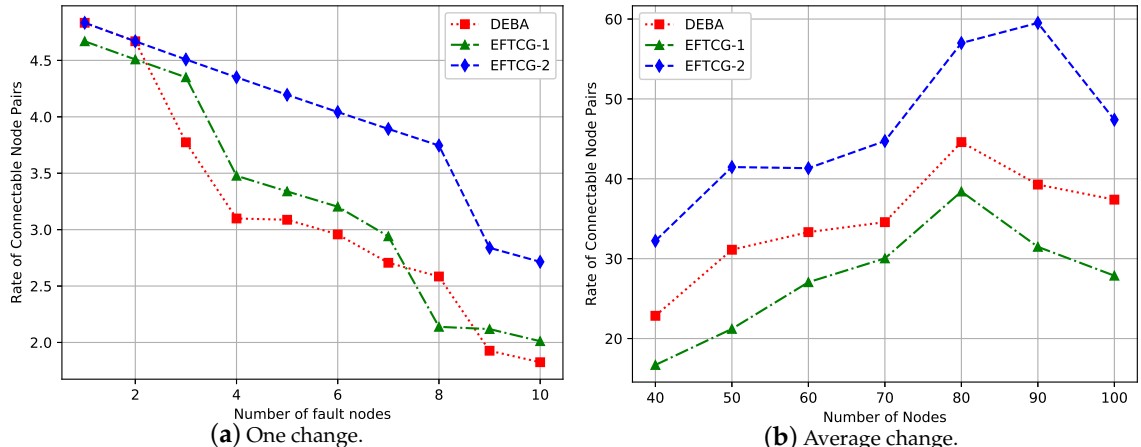

**Figure 13.** The rate of connectable node pairs, (**a**) the change of the rate of connectable node pairs with the generation of failure nodes when the number of nodes is 100; (**b**) the change of the rate of connectable node pairs under different node density after 10 failure nodes appear.

As shown in Figure 14, we compare the network lifetime of EFTCG-1, EFTCG-2, and DEBA-1, which only guarantees the network connectivity, and DEBA-2, which guarantees the network biconnectivity. It can be seen that the network lifetime of EFTCG-2 is shorter than EFTCG-1, but better than DEBA-2. This is because that EFTCG-2 needs to increase the transmit power of nodes to consume more energy, so the network lifetime is shorter than EFTCG-1 algorithm. But It efficiently better balances the factors such as network connectivity, residual energy, and transmit power than DEBA-2, so the network lifetime is longer than DEBA-2.

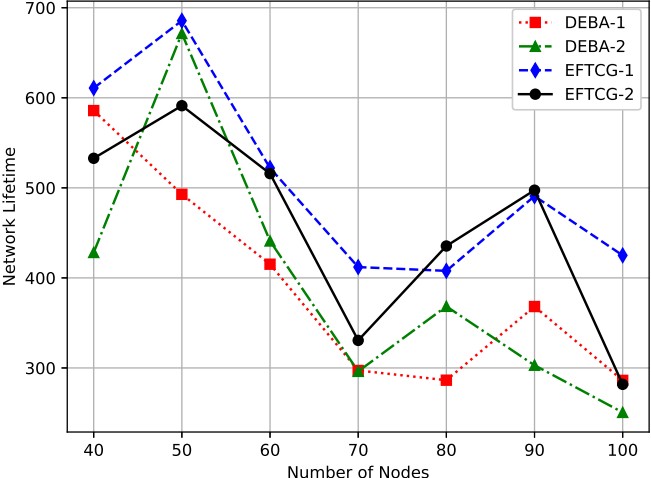

**Figure 14.** The network lifetime.

Table 2 shows the average improvement in fault-tolerant effect under different connectivity when the number of nodes ranges from 40 to 100. Table 3 shows the average reduction of energy-saving effect. The increase rate and decrease rate in Tables 2 and 3 are both EFTCG-2 to EFTCG-1. Through the data analysis of these two tables, it can be concluded that EFTCG-2 can effectively improve the fault-tolerant ability of a network, increasing the robustness of network when lifetime is appropriately reduced.

**Table 2.** The average improvement of fault tolerance.

| Performance | EFTCG-1 | EFTCG-2 | Increase Rate |
|---|---|---|---|
| The average node degree | 2.05 | 3.29 | 60.49% |
| The rate of survival nodes | 45.71% | 62.32% | 36.34% |
| The rate of connectable node pairs | 27.43% | 46.55% | 69.70% |

**Table 3.** The average reduction of energy saving.

| Performance | EFTCG-1 | EFTCG-2 | Decrease Rate |
|---|---|---|---|
| Lifetime | 410 | 392 | 4.4% |

## 6. Conclusions

Nodes in wireless sensor networks have limited battery capacity. This forces the nodes to conserve their energy consumption. At the same time, if the application environment is complex and harsh, the network needs a certain fault-tolerant ability. Thus, a topology control model based on the ordinal potential game is constructed. By considering network connectivity, power, and energy synthetically, we design a practical utility function which achieves the goal of both energy-efficient and fault-tolerant, then propose the EFTCG algorithm. The two subalgorithms of EFTCG algorithm proposed in this paper obtains remarkable results: EFTCG-1 has better energy-efficient effect than other game-based algorithms, and it can effectively reduce the transmit power and prolong the network lifetime. EFTCG-2 has pretty good fault-tolerant ability, and it improves the robustness and reliability of the network when network lifetime is appropriately reduced. The experimental results show the validity of the designed utility function. It can effectively balance transmit power, residual energy, and network connectivity to achieve adequate network energy efficiency and fault tolerance.

In the research process of work, it is found that the number of key nodes which are one-hop or multi-hop neighbor nodes of the sink node nodes is closely related to the network lifetime. Hence, we will establish an appropriate topology control game model, for example, Stackelberg game, to analyze the game of key nodes and ordinary nodes in the future work.

**Author Contributions:** Conceptualization, Y.D. and J.X.; software, Y.D.; supervision, Y.D.; validation, X.H.; writing—original draft, Y.D.; writing—review and editing, J.G.

**Funding:** This work was partially supported by the National Natural Science Foundation of China (11461038), and the Natural Science Foundation of Gansu Province(144NKCA040).

**Conflicts of Interest:** The authors declare no conflicts of interest.

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
