# Peer review of "An Energy-Efficient and Fault-Tolerant Topology Control Game Algorithm for Wireless Sensor Network"

_electronics, doi:10.3390/electronics8091009_

Round 1
Reviewer 1 Report
The authors try to improve the existing algorithm for topology control in WSN based on game theory. Author extended existing formula (novelty) with adding alfa and beta coefficients that reflect current energy rate on the node and evaluating fault tolerance by exploring performance of algorithm with different levels of connectivity k=1 and k=2 for proposed EFTCG-2 algorithm.
Text is constructed in accordance with scientific requirements. All section are presented. But results dont carry out high scientific significance because DEBA, DIA and MLPT algorithms have been already compared in [16] and in the previous work of the same authors (2018) so the article partially repeats the results obtained in that articles (or some results are even contradict!). It would be more interesting to see the comparison with, for example DEBA algorithm as prototype of game theory based algorithm, and other proposed/existing TC algorithms like LEACH, CDS based (connected dominating set) A1, A3 algorithms, or Learning automata based algorithm LECT.
Line 27 - "Topology control is a key technology of WSN". TC is not technology, it is mechanism, way, solution.
Line 74, 112: spelling or grammar error
Overlapping of variables in network model (section 2.1) and game theory (section 3.1): u is cut point and utility function at the same time, p is power and strategies at the same time…
Described network model is not used further in the article. It seems reasonable to remove this section at all.
Figures: on axis x and y measurement units are not specified (sec, hours…)
You use Network running unit as measurement time unit for simulation. Explain what it does mean.
Small remark (no need to change) about lifetime: authors uses the number of network running round when the first node dies as the network lifetime value. But usually, the network lifetime is the number of periods that the network survives until can’t construct a backbone for the network (or active nodes remain connected) that means that nodes could be out of service (died or corrupted) but connectivity still exists.
Reviewer 2 Report
You have a similar paper paper in MDPI Sensors. I would welcome a short statement on how this manuscript fits into the greater scheme of your (obviously ongoing) work, This ultimately relates to the last comment regarding the conclusion where your Future Work is simply to do more work. It seems that you are pursuing the work presented here in the context of a larger investigation and I think it would benefit the paper if you placed this work in the context of that work. Pervious work has shown X, here we show Y (which is relevant and different than X for the following reasons) and in the future we will continue working on this by focusing on the following aspects, see Future work for details.
many occurrences: what is an ordinal game? What is a Nash Equilibrium? Generally, if something appears once, explain it in a quick sentence, if you use a concept often introduce it in the introduction / background section.
[-] With regard to Ordinal Games: You dive into this with a theorem ... the term should first be defined (please do this also in natural language to spare the reader the working through the definition to understand the term) before you postulate / prove the theorem.
A quick search on MDPI returned this special issue, maybe there are some citable references in this? https://www.mdpi.com/journal/sensors/special_issues/Emerging_Sensor_Networks
[-] With regard to Nash Equilibrium: Definition 4 is very hard to understand unless the concept is explained first; there is space and opportunity to casually introduce this notion earlier and then define the part of it that you really need formally in Def 4. See comments below for potential references you can use
References in MDPI journals (its nice to reference MDPI) you could use (pick your preferred option):
https://www.mdpi.com/2414-4088/2/4/63
https://www.mdpi.com/2073-4336/9/1/8
https://www.mdpi.com/2071-1050/11/7/1983
https://www.mdpi.com/2073-4336/9/3/39
line 49: please explain to the uninitiated reader what a Nash-Equilibrium means in the context of topology for WSN
line 51: what does executing nodes in different order mean?
line 92/93: where is the difference between node power and node maximum power? Is p_i at a specific moment of time? If this is generally the maximum power, why is there a subscript i? In line 106 you state just that as assumption 1.
line 107: what is the definition of a sink node?
line 116: I disagree with the claim that energy-efficient topologies imply that node failure is higher than otherwise. Network division is (because you have more cut points) but the nodes should not individually be affected by the topology design.
line 124: the division causes the components to be disconnected, not connected
line 125: node failure very much affects the communication between nodes, it does not affect the overall connectivity of the WSN but e.g. the nodes to the left adn right of B1 cannot communicate with one hop anymore.
line 135: I would argue that this sentence sums up the entire preceding paragraph (line 110 to 135): redundancy implies just that ... removing a link is inconsequential if that link was redundant. (This does not mean you should remove this paragraph, but consider what you are trying to say and whether you need this much text to say it). You state in line 137 that you want to ensure that there are at least two connections that have to be removed to divide a network. Is that all you are trying to state here?
line 160: define sub-tree ... its rather relevant here. In a spanning tree any node with more than two outgoing edges is a cut node ... I find this explanation lacking (its clear to someone who already knows what you are trying to say, but then ... why say it in the first place?). Write as if the reader is not familiar with what you are trying to point out.
line 155: do not number the equation. You do not use the number and in the current state you have two versions of (1). Also, remove the comma
generally: there is a huge amount of work on graph theory out there. It might be useful to refer to introductory / advanced literature and then simply write in clear words (instead of formulae) what you are doing. My reasoning is that the Graph Theory fan will already know this, and the uninitiated reader will be better of to have you tell him/her in clear words what you do and then refer to the literature. But thats just a suggestion ... use your own judgement here
line 165-170: you already use p above for power, do not reuse the same letter here (why not use \sigma?). Also ... you write that s_i is the strategy (but you just defined s to denote strategy profiles). I would also propose to write s_{N\{i}} instead of s_{\_i} for readability but thats up to you. Generally speaking, the notation can be improved (use A for agents instead of players, or use e or \epsilon for the power (energy) in the paragraph above). Again, use your own judgement.
line 171-173: does the utility of a strategy not depend on the state / outcome? What type of game are you talking about in the first place? Is this perfect/imperfect information? I guess this simplified / shortened definition could work but you at least have to explain what a game is a bit better here. Use your own judgement and do not add the entire theory of game theory to the paper, instead, add a paragraph that explains how your application is a game (make it a bit more formal than what you have and relate it to the definition you provide here).
Generally, Section 3.1 could be improved. Suitable references in MDPI journals (its nice to reference MDPI) you could use (pick your preferred option / find your own preferred literature (do not use your own paper in Sensors, your definitions there are equally unsatisfying)):
https://www.mdpi.com/2414-4088/2/4/63
https://www.mdpi.com/2073-4336/9/1/8
https://www.mdpi.com/2071-1050/11/7/1983
https://www.mdpi.com/2073-4336/9/3/39
and many more
line 187-194: try and rephrase this paragraph ... do not be shy to write a few more sentences to explain what you try to use. I feel that understanding this is rather relevant for the reader and its not easy the way you have written this (the English does not help). It would also help if you define the terms "power efficiency" and what you mean by energy distribution (both can happen at an earlier stage in the paper). The same holds for "residual power" (used thereafter in your manuscript). Power efficiency can be defined in terms of power consumption, then point out that "the lower that value the higher the efficiency" / can be defined against the ideal value of consumption in a specific network.
line 209: this is not obvious to me but (as I will point out below) some of the math is not clear to me in the first place so this might be obvious to the expert reader. Use your own judgement.
line 222: how do nodes "select" neighbours
line 224: why is the order of magnitude different? Where does this come into play / where is this defined? I feel that this is a relevant point to clarify because you argue that "because of this [...] cannot be effectively compared". I am also wondering whether this is true (whether it cannot be compared) because you then continue to do just that .... using normalization etc.
line 226: You reference the weights in other algorithms ... are these algorithms in published literature you already reference? If so then please add the references here. If not then please add 2-3 references here (please pick papers that themselves have received citations not some obscure publications). Furthermore ... you "think"? I like that you add your own view but please motivate this / support your suggestions.
line 229: its better to say that that alpha (and thus beta) are determined by the residual energy (which you have thus far not defined) instead of attributing agency to the nodes.
line 230: I would paraphrase this sentence. I like that you explain the impact of large and small values of the residual energy so please keep this, just paraphrase it. Consider adding a sentence on how the resulting behaviour in turn affects the residual energy of a node (and, if this is applicable, the effect this has on the node's neighbours).
line 233-249: I (have to) believe your proof ... because (a) I cannot be asked to work through it myself and (b) its very (!) hard to work through without any accompanying explanation. Please add to this section ... a short paragraph here and there (maybe labelled "Explanation") would greatly add to this. Explain to the reader what you have done (so the interested reader can go through your proof) and do so in a way that makes it understandable what your proof does. In the end, most readers would strongly prefer to get a flavor of the proof and skip the actual math works. This makes your paper significantly better and will get you the good will of the reader (and, evidently, the reviewers). Consider that there should be a reason why you have added section 3.3 in the first place ... if the reader can understand this and the content of the section by reading the explanation you have improved the value of your paper.
Section 3.3.: I leave it at this and postpone the checking of the math until the revised version is submitted.
line 252-253: These sentences are not good English. Furthermore, since this basically refers to the following sections just use \S \ref{} instead of (1) and (2).
line 267-275: This is very hard to understand again. You use terms that are not yet defined, the math is very hard to read and the context of the Nash Equilibrium is not clear.
Algorithm 1 shows both phases so line 275 incorrectly states that Alg 1 shows the topology game phase. NE means Nash Equilibrium but please introduce abbreviations before you use them.
line 278: simulations were conducted using a python [implementation]. How many runs, what was the cut-off ... I am missing the methodology description here.
Gnerally I would like to first have a section in which tests you did, and why. What is the motivation for the different test? Once that is in place you can start telling me how you collected the data and then show me the results (and discuss them).
line 283: since the three algorithms were mentioned and referenced before, add the references here again please.
lines 285-... this is a discussion of the results. Some of the claims are hard to follow ("... may die early"), please add to this.
Generally, try to separate the results and the discussion thereof.
Figure 5: What is "number of network running around"?
line 300: You claim to have varied the number of nodes ... how? in steps of 10? in steps of 5? Why? Is there a preliminary test that helped you make this choice (whatever it was)? If yes, please tell me about it, if not you should at least add a sentence indicating that this was not a bad choice. The same holds for your decision to run 10 simulations ... why not 100? (10 is totally okay, but I'd like to know what motivated the choice). Can you add a sentence telling the reader how long these simulations took? This does not have to be to the millisecond but I'd like an idea about execution time.
Figures 4 and 10: the numbering is unnecessary (no need to change it unless you have this readily available). If available you could state properties of the topologies in the caption (such as number of crucial nodes, average connectivity factor etc), that way the networks can be meaningfully compared. Again ... if this is not readily available you do not need to go back and add this, but if you have it it would add to the Figures.
For Figures 5-9, 11-14: generally I would suggest to make the captions of the figures far more explanatory. Consider that a casual reader should be able to just look at a figure and get an idea of what he/she is looking at by reading the caption. This is a good place to add a simplified natural language explanation to your section. (use your own judgement)
line 307-312: Some claims are made here that are not backed up. I would also like to have some discussion as to WHY the results are the way they are.
line 313-318: see above
line 319-326: see above
Table 2: this is relevant and interesting information but its deserves a paragraph to (a) explain the relevance and (b) how you come by these values. Natural language suffices for both (a) and (b).
Section 5.2: why do you not include the other algorithms in the analysis? Again, motivating your choice suffices but you need to provide me with a justification. As above, I am also missing the methodology and the justification for your choices (random? Since you have insight into which nodes are more crucial to the network performance, why did you not focus on these nodes?). You state that you used the value 10 for your fault nodes ... was that the same for all network sizes? Why did you not chose a percentage of the nodes in the network or relate this to the number of crucial nodes?
line 335-345: I am skipping the math here but these definitions are good: you define your performance measure before you use it. I would have liked this for the previous section as well.
line 345-348, 249-255, 356-362, 363-366, 367-370 and 371-375: see above. These paragraphs tell me (a) what you chose for the simulation (but not why) and otherwise narrate the graphs (but do not discuss why they are the way they are, nor what this implies about the algorithms).
Section 6: this can be extended. You did good work here but the conclusion seems like an afterthought. Please extend on this ... and the last sentence "the next step is do to more work" is meaningless: what work, and why? What does your paper not address? I would also like to hear how this will benefit the field ... practically (use your own judgement here).
===========================================================
typos (examples only, please check the entire document)
general comment: do not use "can't", instead use "cannot" (and while "can not" is acceptable, "cannot" is preferable; at least you should only use one of the two consistently)
general comment: [of] [the] [a] etc are missing in many places
line 52: guarantee[s], ensure[s]
line 56: both [of] them
line 70: based on [the] ordinal potential game
line 71: the existence of [a] Nash Equilibrium
line 81: failure nodes --> node failure ??
line 112: [N]ode failure
line 167: strateg[ies]
Round 2
Reviewer 1 Report
Taking into account the work done by the authors to eliminate observations I can recommend this paper for publication.
Reviewer 2 Report
There are minor corrections regarding the presentation and the English, but these are really minor and I assume that the editorial staff will address these in the final round of copy-editing the paper.
As far as I can see you have addressed the comments I had, thank you and best regards.